# Factors affecting chronic low back pain among high school baseball players in Japan: A pilot study

Hidetoshi Nakao[1]*, Ryota Imai[2], Taro Hamada[3], Masakazu Imaoka[2],
Mitsumasa Hida[2], Takeshi Morifuji[1], Masashi Hashimoto[4]

1 Department of Physical Therapy, Faculty of Social Work Studies, Josai International University, Togane City, Chiba, Japan, 2 School of Rehabilitation, Osaka Kawasaki Rehabilitation University, Kaizuka City, Osaka, Japan, 3 Department of Rehabilitation, Osaka Global Orthopedic Hospital, Joto-ku Osaka City, Osaka, Japan, 4 Department of Rehabilitation, Faculty of Health Sciences, Nara Gakuen University, Nara City, Nara, Japan

☯ These authors contributed equally to this work.
* h_nakao@jiu.ac.jp

**Data Availability Statement:** The data are available on Figshare (DOI: 10.6084/m9.figshare. 20097353).

**Funding:** The author(s) received no specific funding for this work.

## Abstract

The prevalence of chronic lower back pain (CLBP) among baseball players is high. CLBP is associated with reduced participation in practice and games. This pilot study examined the factors associated with CLBP among high school baseball players in Fukui, Japan. The participants underwent two health examinations in high school: (1) as first-grade baseball players (baseline) and (2) as second-grade baseball players (follow-up); a total of 59 players who could be followed-up a year later were included in the study. Players were divided into three groups based on whether they had no lower back pain (LBP) (n = 30), improved LBP (n = 17), or CLBP (n = 12) after 1 year of follow-up. Players were evaluated on the physical and cognitive aspects of pain. The Number Rating System, Pain Catastrophizing Scale (PCS), Tampa Scale for Kinesiophobia (TSK), Central Sensitization Inventory (CSI), body characteristics (age, height, weight, body mass index, and skeletal mass index), and a medical history questionnaire regarding spondylolysis and baseball loads were used to evaluate the players. Inventory scores were highest in the CLBP group, which indicated that this group had significant pain that affected their willingness to engage in baseball-related activities. The TSK scores in the CLBP group were worse on follow-up. High school baseball players with CLBP were more likely to have lumbar spondylolysis and kinesiophobia, which are also factors related to pain chronicity. Kinesiophobia and the presence of lumbar spondylolysis should be considered when creating an exercise program for high school baseball players with CLBP.

## Introduction

Baseball has a large following in Japan, and lower back pain (LBP) is a significant problem among baseball players because it reduces participation time and increases the risk for disability; LBP occurs not only in adulthood but also among young athletes [1]. The LBP prevalence

**Competing interests:** The authors have declared that no competing interests exist.

among young baseball players aged 12–15.5 years) ranges from 8.3–15% [2]. Furthermore, LBP affects up to 48% of Japanese college baseball players [3], and the prevalence is also high in older baseball players. Therefore, it is important to analyze the pain-related factors associated with chronic LBP (CLBP) in high school players so that appropriate measures can be implemented.

LBP can interfere with athletes' performance and in some cases, prevent them from playing the game [4]. Several factors, such as age, gender, obesity, and inactivity, have been reported to be associated with LBP in the general population [5–7]. Obesity is the most important factor associated with CLBP and should be considered in athletes [8]. In addition, baseball players routinely experience high torsional and rotational forces on the lumbar spine during hitting and throwing. This motion during baseball can lead to back stiffness, sacroiliac joint pain, or discogenic or facet joint pain [9]. We hypothesized that examining several related factors such as pain intensity, physical composition, and baseball-related load in high school baseball players with CLBP may help implement corrective measures for improving LBP.

Chronic pain constitutes both physical and mental aspects. The mental aspects include depression, anxiety, catastrophizing, kinesiophobia, and a decline in self-efficacy [10–14]. In addition, reports have identified psychosocial factors and central sensitization as risk factors for CLBP [15, 16]. Psychosocial factors may hinder athletes from resuming the sport and affect their performance.

To the best of our knowledge, no study has examined the association of body compositions with CLBP intensity in high school baseball players. Hence, this study aimed to examine factors, such as physical composition, pain evaluation, and load related to LBP, that contribute to CLBP among high school baseball players. We also aimed to identify factors that promote and alleviate CLBP.

## Materials and methods

### Study design and population

This study was conducted in accordance with the Declaration of Helsinki and approved by the Osaka Kawasaki Rehabilitation University Research Ethics Review Board (approval number: OKRU-RA0003). Furthermore, written informed consent was obtained from each participant.

Baseball players from a high school in Fukui Prefecture, Japan, were included in this study. Only players who could participate in practices despite having LBP were analyzed. Players with severe physical or mental conditions that prevented participation in games or with a history of lumbar surgery or central nervous system disorders were excluded. The practice frequency of the players was 6 days a week with an average of 4–5 hours per day. Japanese high school baseball games are conducted during the off-season period between December and March, and there are no external games. The practice activities during this period are running, defensive and batting, and strength training. The number of pitching practices decreased at this time.

Players completed a health survey conducted from December 2019 to December 2021. The questionnaire evaluated pain intensity, baseball-related movement, and overall performance. Based on the answers to the questionnaire, the players were grouped into the CLBP (n = 12), improved LBP (n = 17), and no LBP (n = 30) groups. Players in the CLBP group demonstrated persistent LBP between grades 1 and 2. Players in the improved LBP group responded to the questionnaire with no LBP when implemented a year later. Furthermore, players in the no-LBP group demonstrated no pain in either survey.

### Pain intensity

We used a questionnaire to measure pain intensity and its psychological impacts. Pain was defined as the maximum pain felt at a specific site and scored according to a numerical

rating scale (NRS) from 0 ("no pain") to 10 ("the worst imaginable pain"). CLBP was defined as persistent pain for at least ≥12 weeks and pain intensity of cause by a visual analog scale of ≥10 mm [17].

We also administered the Pain Catastrophizing Scale (PCS) [18], the shortened Japanese version of the Tampa Scale for Kinesiophobia (TSK) [19], and the Central Sensitization Inventory (CSI) [20, 21]. The PCS comprised 13 items in three domains: rumination (five items), helplessness (five items), and magnification (three items). Each item is rated on a five-point scale from 0 ("not at all") to 4 ("all the time"). The total PCS score ranged from 0 to 52. High scores indicate greater pain catastrophizing. TSK comprised 17 questions for assessing pain-related motor kinesiophobia; however, this study used a shortened version with 11 questions. TSK is used worldwide and has been validated. The CSI was developed as a screening tool to identify and quantify patients with central sensitization (CS) related symptoms. CSI-9, a shortened version of CSI, is a nine-item symptomatological and self-reported questionnaire that assesses common health-related symptoms in individuals with central susceptibility syndrome [13, 22]. The effectiveness of CSI as an assessment tool for patients with chronic pain has been demonstrated, and the total CSI score is associated with a wide range of pain, pain intensity, disability, quality of life (QOL), and pain catastrophe [23].

## Body composition

Physiological parameters were measured using bioelectrical impedance analysis (Inbody, Tokyo, Japan) at 20 and 1000 kHz frequencies. In addition, data were obtained from the participants' electronic medical records [24]. Body mass index (BMI) was calculated by dividing body weight (kg) by the square of the player's height ($m^2$), whereas the appendicular skeletal muscle mass index (SMI) was derived from appendicular muscle mass (kg) divided by the square of the player's height ($m^2$).

## Baseball-related loads and injury

We evaluated the presence of pain during baseball-related movements, such as batting, throwing, ball-catching, running, and performing barbell squats. Running in baseball means a movement involving a high load, such as a dash. We collected data on the lumbar disc herniation and/or lumbar spondylolisthesis through a thorough medical history. This data was collected using a question in a Japanese language questionnaire: "Have you ever been diagnosed with the lumbar disc herniation or lumbar spondylolysis?"

## Sample size

This pilot study was conducted to estimate the sample size required to identify differences among the three groups. Using IBM SPSS version 27 (IBM Corp., Armonk, NY, USA), the study was implemented to estimate the sample size, considering the mean difference among no LBP, improved LBP, and CLBP groups of 9.4 ± 5.7, 12.8 ± 4.7, and 11.7 ± 4.7, respectively for TSK outcomes. Further, 12 participants were required per group, with a statistical power of 0.8, for comparing groups using the Kruskal−Wallis test. The validity of the sample size measurement was based on the recommendation to employ approximately 12−15 participants per group in a pilot study [25].

## Statistical analysis

Data were recorded and analyzed using IBM SPSS version 27. The results are presented as means and standard deviations. The Shapiro−Wilk test was used to confirm normal distribution

at each endpoint. Age and PCS scores were compared using the analysis of variance test, whereas the Kruskal–Wallis test was used to analyze the height, weight, BMI, SMI, TSK, rumination, helplessness, magnification, CSI, and NRS. Post-hoc comparisons were made using the Dunn test. Fisher's exact test was used to analyze the lumbar disc herniation and lumbar spondylolysis, pain during baseball-related movement, and muscle function. A subgroup analysis was performed in the CLBP group to identify the factors that contributed to chronicity. A Wilcoxon signed-rank test was used to analyze pain intensity, and the McNemar test was performed for baseball-related pain (running and barbel squat) at baseline and follow-up.

## Results

Fig 1 outlines the selection of study participants. A total of 69 high school baseball players were examined. Eight players who had no LBP in the first grade but showed LBP in the second grade and two players who had incomplete questionnaire data were excluded. Table 1 shows the baseball experience, position, presence or absence of LBP, and LBP period at baseline. A total 59 participants were included in this study, with a mean age and baseball experience of 15.8 and 9.5 years, respectively.

### Demographic characteristics and pain evaluation among the groups

The demographic body composition and pain intensity of all three groups are summarized in Table 2. Based on the data collected during the follow-up, 12 (20.3%), 17 (28.8%), and 30 (50.8%) players had CLBP, improved LBP, and no LBP, respectively.

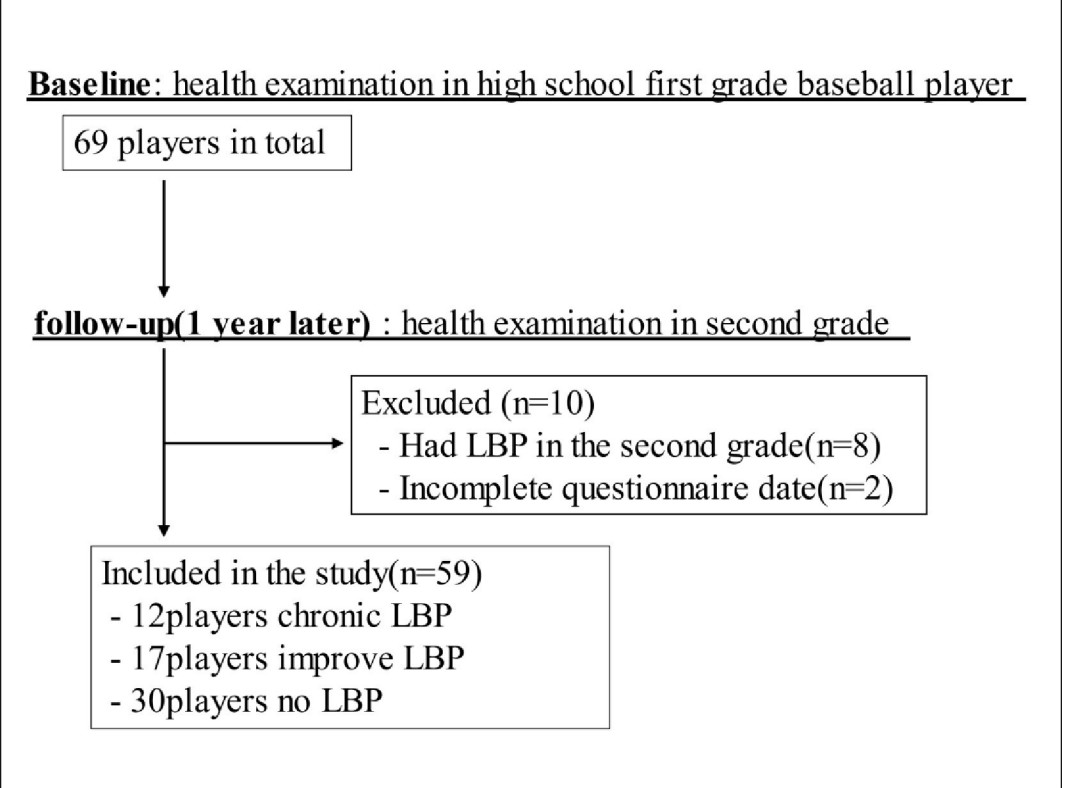

**Fig 1. Flowchart of the participants' selection.**

**Table 1. Baseline characteristics of the participants.**

| | Total (n = 59) | | NLBP (n = 30) | | ILBP (n = 17) | | CLBP (n = 12) | |
|---|---|---|---|---|---|---|---|---|
| | Mean (SD) | n(%) | Mean (SD) | n(%) | Mean (SD) | n(%) | Mean (SD) | n(%) |
| **Age(year)** | 15.8 (0.3) | | 15.7 (0.4) | | 15.9 (0.2) | | 15.8 (0.3) | |
| **Baseball experience (year)** | 9.5 (1.4) | | 9.6 (1.4) | | 9.5 (1.4) | | 9.3 (1.5) | |
| **Position** | | | | | | | | |
| Picher | | 11 (19.0) | | 5(16.7) | | 1 (6.2) | | 5 (41.7) |
| Non-picher | | 47 (81.0) | | 25(83.3) | | 15 (93.8) | | 7 (58.3) |
| **LBP period,** | | | | | | | | |
| < 12week | | 25 (86.2) | | 0(0) | | 15 (88.2) | | 10 (83.4) |
| 12week | | 4 (13.8) | | 0(0) | | 2 (11.8) | | 2 (16.6) |

**Table 2. Comparison of body composition and pain intensity among the three groups.**

| Characteristics | | | | | | |
|---|---|---|---|---|---|---|
| **Baseline** | **Total** (n = 59) | **NLBP** (n = 30) | **ILBP** (n = 17) | **CLBP** (n = 12) | **Kruskal–Wallis** | **Dunn test** |
| | | a group | b group | c group | | |
| **Body composition** | | | | | | |
| Height (cm) | 171.1 (0.3) | 170.6 (6.2) | 173.6 (5.3) | 169.6 (4.6) | n.s. | |
| Weight (kg) | 65.3 (8.3) | 64.6 (9.2) | 67.4 (7.5) | 64.8 (6.6) | n.s. | |
| BMI (kg/m2) | 22.2 (1.9) | 22.1 (2.1) | 22.2 (1.9) | 22.2 (1.9) | n.s. | |
| SMI (kg/m2) | 8.1 (0.5) | 8.1 (0.6) | 8.1 (0.5) | 7.9 (0.5) | n.s. | |
| **Pain measurement** | | | | | | |
| TSK | 10.6 (5.3) | 9.4 (5.7) | 12.8 (4.7) | 11.7 (4.7) | n.s. | |
| PCS total score | 17.5 (11.6) | 13.3 (12.1) | 19.7 (19.7) | 27.5 (9.2) | <0.05 | a-c(*) |
| Rumination | 10.6 (5.9) | 8.0 (6.1) | 12.3 (3.9) | 15.4 (3.5) | <0.05 | a-c(*) |
| Helplessness | 3.0 (3.6) | 2.6 (3.2) | 4.1 (2.7) | 6.6 (2.2) | n.s. | |
| Magnification | 3.7 (3.2) | 2.6 (3.2) | 4.1 (5.4) | 6.6 (2.2) | <0.05 | a-c(*) |
| CSI | 21.0 (5.7) | 18.6 (5.0) | 21.8 (4.0) | 26.4 (5.7) | <0.05 | a-c(*) |
| NRS | 1.72 (2.1) | 0.5 (1.3) | 2.5 (1.9) | 3.3 (2.6) | <0.05 | a-b(*) a-c(*) |
| **Follow-up** | Total (n = 59) | NLBP (n = 30) | ILBP (n = 17) | CLBP (n = 12) | Kruskal-Wallis | Dunn test |
| | | a group | b group | c group | | |
| **Body composition** | | | | | | |
| Height (cm) | 172.4 (5.4) | 170.6 (5.8) | 173.6 (5.1) | 170.1 (4.1) | n.s. | |
| Weight (kg) | 69.1 (8.4) | 67.2 (9.2) | 72.0 (9.4) | 69.7 (6.7) | n.s. | |
| BMI (kg/m2) | 23.4 (2.0) | 23.0 (2.2) | 23.8 (1.5) | 24.0 (1.9) | n.s. | |
| SMI (kg/m2) | 8.4 (0.6) | 8.3 (0.7) | 8.6 (0.4) | 8.5 (0.5) | n.s. | |
| **Pain intensity** | | | | | | |
| TSK | 16.4 (5.8) | 15.1 (5.3) | 15.4 (4.5) | 21.1 (6.3) | <0.05 | a-c(*) |
| PCS total score | 13.1 (12.0) | 11.5 (10.8) | 12.5 (14.1) | 17.8 (11.3) | n.s. | |
| Rumination | 8.4 (6.8) | 7.8 (6.2) | 7.0 (7.3) | 11.6 (6.9) | <0.05 | |
| Helplessness | 2.3 (3.6) | 1.9 (3.2) | 3.0 (4.3) | 2.5 (3.6) | n.s. | |
| Magnification | 2.3 (2.5) | 1.7 (2.1) | 2.5 (2.8) | 3.5 (2.7) | <0.05 | |
| CSI | 9.1 (5.5) | 6.7 (3.9) | 10.5 (5.9) | 12.9 (5.8) | <0.05 | a-c(*) |
| NRS | 1.1 (1.6) | 0.6 (1.4) | 0.8(1.3) | 3.0(1.5) | <0.05 | a-c(*) b-c(*) |

NLBP: no lower back pain, ILBP: improvement lower back pain, CLBP: chronic lower back pain, BMI: body mass index, SMI: skeletal muscle mass index, TSK:Tampa Scale for Kinesiophobia, PCS:Pain Catastrophizing Scale, CSI: Central Sensitization Inventory, NRS: Numeric Rating Scale, n.s.: non-significant

PCS, rumination, magnification, and CSI were significantly lower at baseline in the no LBP group than in the CLBP group. For the NRS, the mean value was lowest at 0.5 points in the no LBP group. Conversely, the improved LBP and CLBP groups scored 2.5 and 3.3 points, respectively, showing a significant difference from no LBP group. At follow-up, TSK and CSI were significantly lower in the no LBP group than in the CLBP group. For the NRS, the mean value of the CLBP group was high (3.0 points), whereas the improved LBP and no LBP groups scored 0.8 and 0.6 points, respectively. The CLBP group had a mean TSK score of 21.1 points, whereas the improved LBP and no LBP groups scored 15.4 and 15.1 points, respectively. The CLBP group had a mean CSI score of 12.9 points, whereas the improved LBP and no LBP groups scored 10.5 and 6.7 points, respectively.

The CLBP group was also significantly more likely to have a history of lumbar disc herniation and/or spondylolysis, pain on baseball-related movement, and poorer muscle function (Table 3). Pain was experienced mostly during running and barbell squats.

### Pain evaluation in the CLBP group

Table 4 shows the pain evaluation scores of the CLBP group at baseline and follow-up. No significant differences were observed in pain intensity and helplessness; however, TSK scores were significantly higher at follow-up than at baseline ($p < 0.05$). In contrast, PCS and CSI scores, rumination, and magnification were significantly lower at follow-up than at baseline ($p < 0.05$). Table 4 demonstrates the incidence of pain during baseball-related movement in the CLBP group. No significant differences were observed in LBP presence during running and barbell squats.

### Discussion

In this study, we examined the incidence and factors associated with CLBP among high school baseball players at baseline and after 1 year. At baseline, 49.1% of the players experienced LBP, and by the follow-up date, 20.3% developed CLBP. LBP increases with age [26] and is more likely to recur in athletes; this study reports an LBP prevalence of 8.3–15% and 49.1% in young and high school baseball players, respectively [2, 3].

The CLBP group had significantly higher NRS and TSK scores at baseline. On follow-up, the TSK scores had worsened; however, the PCS and CSI scores had improved despite no change in the reported pain intensity.

A previous report examined patients with non-specific LBP for at least 6 weeks. The average TSK score in this population exceeded the cut-off value, which signified a correlation between pain intensity and the Oswestry Disability Index score [27]. Thus, kinesiophobia affected the perceived pain intensity and ability to participate in activities of daily living and sports.

The CLBP group was more likely to have lumbar spondylolysis. Lumbar spondylolysis results from stress fractures of the *pars interarticularis* [28] and is the reported cause of LBP in almost 50% of adolescent athletes [29]. In addition, 30% of Japanese professional baseball and soccer athletes are reported to have lumbar spondylolysis [30]. Thus, our study demonstrated that players with lumbar spondylolysis were more likely to have LBP-associated kinesiophobia, limiting their movement and making CLBP more likely.

Among the baseball-related movements, running and barbell squats were more associated with LBP in the CLBP group. When the baseline and follow-up data were compared, a small reduction in pain during running and barbell squats was observed; however, the difference was not significant. Pain during baseball-related movements has traditionally been associated with swinging and throwing motions because the movements apply high torsional and rotational forces to the lumbar spine [9, 31]. However, our study identified running and barbell squat movements as the most likely triggers for pain, particularly among players with lumbar

**Table 3. Fisher's exact test showed significant differences among the three groups.**

| Baseline | | | | |
|---|---|---|---|---|
| **Evaluation Item** | **NLBP (n = 30)** | **ILBP (n = 17)** | **CLBP (n = 12)** | |
| | **n (%)** | **n (%)** | **n (%)** | **p-value** |
| **Baseline** | | | | |
| **Past medical history** | | | | |
| Lumbar disc herniation | | | | |
| Presence | 0 (0) | 0 (0) | 1 (8.3) | 0.632 |
| Absence | 30 (100) | 17 (100) | 11 (91.7) | |
| Lumbar Spondylolysis | | | | |
| Presence | 4 (13.3.) | 1 (5.9) | 6 (50.0) | 0.002 |
| Absence | 26 (86.7) | 16 (94.1) | 6 (50.0) | |
| **LBP related loads** | | | | |
| Batting | | | | |
| Presence | 0 (0.0) | 4 (23.5) | 6 (50.0) | <0.001 |
| Absence | 30 (100.0) | 13 (76.5) | 6 (50.0) | |
| Throwing | | | | |
| Presence | 0 (0.0) | 1 (5.9) | 5 (41.7) | <0.001 |
| Absence | 30 (100.0) | 16 (94.1) | 7 (58.3) | |
| Ball catching | | | | |
| Presence | 0 (0.0) | 0 (0.0) | 2 (16.7) | 0.017 |
| Absence | 30 (100.0) | 17 (100) | 10 (83.3) | |
| Running | | | | |
| Presence | 0 (0.0) | 6 (35.3) | 8 (66.7) | 0.013 |
| Absence | 30 (100.0) | 11 (42.9) | 4 (33.3) | |
| Barbell squat | | | | |
| Presence | 1 (3.3) | 13 (76.5) | 6 (50.0) | <0.001 |
| Absence | 29 (96.7) | 4 (23.5) | 6 (50.0) | |
| **Follow-up** | | | | |
| LBP related loads | | | | |
| Batting | | | | |
| Presence | 0 (0.0) | 0 (0.0) | 2 (16.7) | 0.019 |
| Absence | 30 (100.0) | 17 (100.0) | 10 (83.3) | |
| Throwing | | | | |
| Presence | 0 (0.0) | 0 (0.0) | 2 (16.7) | 0.019 |
| Absence | 30 (100.0) | 17 (0.0) | 10 (83.3) | |
| Ball cathing | | | | |
| Presence | 0 (0.0) | 0 (0.0) | 1 (8.3) | 0.142 |
| Absence | 30 (100.0) | 30 (100.0) | 11 (91.7) | |
| Running | | | | |
| Presence | 0 (0.0) | 0 (0.0) | 5 (13.2) | <0.001 |
| Absence | 30 (100.0) | 17 (0.0) | 7 (41.7) | |
| Barbell squat | | | | |
| Presence | 0 (0.0) | 0 (0.0) | 4 (33.3) | <0.001 |
| Absence | 30 (100.0) | 17 (100.0) | 8 (66.7) | |

NLBP: no lower back pain, ILBP: improved lower back pain, CLBP: chronic lower back pain

Fisher's exact test

**Table 4. A prospective cohort survey of characteristics for CLBP.**

| Characteristics | Baseline (n = 12) | Follow-up (n = 12) | |
|---|---|---|---|
| | Mean (SD) | Mean (SD) | p-value |
| Pain intensity | | | |
| TSK | 11.7 (4.7) | 21.1 (6.3) | p<0.05$^{\alpha}$ |
| PCS total score | 27.5 (9.2) | 17.8 (11.3) | p<0.05$^{\alpha}$ |
| Rumination | 15.4 (3.5) | 11.6 (6.9) | n.s. |
| Helplessness | 6.6 (2.2) | 2.5 (3.6) | n.s. |
| Magnification | 6.6 (2.2) | 3.5 (2.7) | n.s. |
| CSI | 26.4 (5.7) | 12.9 (5.8) | p<0.05$^{\alpha}$ |
| NRS | 3.3 (2.6) | 3.0 (1.5) | n.s. |
| Running | | | |
| Presence | 8 (66.7) | 5 (13.2) | n.s. |
| Absence | 4 (33.3) | 7 (41.7) | |
| Barbell squat | | | |
| Presence | 6 (50.0) | 4 (33.3) | n.s. |
| Absence | 6 (50.0) | 8 (66.7) | |

TSK: Tampa Scale for Kinesiophobia, PCS: Pain Catastrophizing Scale, CSI: Central Sensitization Inventory, NRS: Numeric Rating Scale, n.s.: non-significant

$^{\alpha}$Wilcoxon signed rank test

McNemar test

spondylolysis. While running has not been identified as a risk factor for LBP [32], baseball players with CLBP should demonstrate more care when performing these movements.

Our study suggested that kinesiophobia is significantly related to CLBP among high school baseball players. A study by Zawandka *et al.* [33] demonstrated that squatting requires a greater range of knee and hip joint motion among participants with CLBP, whereas Osumi *et al.* [34] reported that patients with high TSK scores require more time to initiate a movement and return to the original position. Thus, kinesiophobia affects joint movement ability; therefore, continuing training with running and squats poses a risk. However, aerobic exercises such as walking [35] and strength training specific to the thoracic, lumbar, and posterior hip regions have been reported to be effective in improving CLBP [36]. Therefore, an exercise program with a controlled TSK score is recommended for athletes with CLBP.

Our present findings suggest that higher TSK in CLBP is seen among high school baseball players with lumbar spondylolysis. Future studies should investigate the association between kinesiophobia and spondylolysis in other sports, such as racquet and contact sports, as they may also have high TSK scores and lubar spondylolysis rates in CLBP.

This study has some limitations. First, it was a pilot study that analyzed a small number of participants. Second, there is no data on the exercise load. Future studies are needed to determine the relationship between CLBP and exercise load.

## Conclusions

Our study aimed to examine factors, such as physical composition, pain evaluation, and load related to LBP, that contribute to CLBP among high school baseball players. We identified kinesiophobia and lumbar spondylolysis as factors related to CLBP. Therefore, particular attention should be given to kinesiophobia during running and barbell squat training, especially in young athletes. Furthermore, the TSK score and presence of lumbar spondylolysis should be considered when creating an exercise program for high school baseball players with CLBP.

## Acknowledgments

The authors are grateful to the participants involved in this study. The experiments comply with the current laws of Japan where they were performed.

## Author Contributions

**Conceptualization:** Hidetoshi Nakao.

**Data curation:** Hidetoshi Nakao, Ryota Imai.

**Formal analysis:** Hidetoshi Nakao.

**Funding acquisition:** Hidetoshi Nakao.

**Investigation:** Hidetoshi Nakao, Ryota Imai, Masakazu Imaoka, Mitsumasa Hida.

**Methodology:** Hidetoshi Nakao.

**Project administration:** Taro Hamada.

**Supervision:** Masashi Hashimoto.

**Visualization:** Hidetoshi Nakao.

**Writing – original draft:** Hidetoshi Nakao, Takeshi Morifuji.

**Writing – review & editing:** Hidetoshi Nakao, Ryota Imai, Taro Hamada, Masakazu Imaoka, Mitsumasa Hida, Masashi Hashimoto.

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
