## [Decision Letter · Decision Letter 0]

2 May 2022

PONE-D-21-25318

Factors Affecting Chronic Low Back Pain Among High School Baseball Players in Japan: A Pilot Study

PLOS ONE

Dear Dr. Nakao,

Thank you for submitting your manuscript to PLOS ONE. After careful consideration, we feel that it has merit but does not fully meet PLOS ONE’s publication criteria as it currently stands. Therefore, we invite you to submit a revised version of the manuscript that addresses the points raised during the review process.

Please, follow the reviewer suggestions to improve your manuscript.

We look forward to receiving your revised manuscript.

Kind regards,

Matias Noll, Ph.D

Academic Editor

PLOS ONE

Journal Requirements:

2. Please provide additional details regarding participant consent. In the ethics statement in the Methods and online submission information, please ensure that you have specified (1) whether consent was informed and (2) what type you obtained (for instance, written or verbal, and if verbal, how it was documented and witnessed). Your study included minors, so state whether you obtained consent from parents or guardians.

Reviewers' comments:

Reviewer's Responses to Questions

**Comments to the Author**

1. Is the manuscript technically sound, and do the data support the conclusions?

Reviewer #1: Partly

2. Has the statistical analysis been performed appropriately and rigorously? 

Reviewer #1: Yes

3. Have the authors made all data underlying the findings in their manuscript fully available?

Reviewer #1: Yes

4. Is the manuscript presented in an intelligible fashion and written in standard English?

Reviewer #1: Yes

5. Review Comments to the Author

Reviewer #1: Authors did a study on Factors Affecting Chronic Low Back Pain Among High School Baseball Players in Japan: A Pilot Study.

However, I think the study contains many factual errors and needs to be substantially revised to be considered for publication.

The first concern I have, regards the language use. I think the study needs substantial English language editing to make it more clear for the readers. Secondly, the authors need to tell us about the importance of their study and findings they reached.

Introduction

The authors start introduction with the sentence IN LINE 45 : „ whereas chronic LBP (CLBP), defined as LBP lasting at least three months, is observed in 1–40% of all baseball players.“ it is needed to be rechecked.

Some sentences need to be more clear for the reader such as in line 51: In addition, reports have identified psychosocial factors and central sensitization as risk factors for CLBP [3, 4]; what does mean…

Sometimes authors present us with statements that need to be followed with references, but the references are omitted. For example, on page …:

The mental aspects of chronic pain

50 include depression, anxiety, catastrophizing, kinesiophobia, and a decline in self efficacy. What literature are the authors referring to?

The introduction section more or less provides hints to the relevant background but I cannot see how the experimental questions can be drawn from the given information. For instance, in line 59 the statement of the problem has finished with this sentence, Preventing or rehabilitating LBP in high school players may reduce LBP prevalence among college and professional athletes”

The hypotheses cannot be drawn from the information provided? I think it is needed to be revised in a standard manner, because it is not the aim of the study.

METHODS

We see very important inconsistency: in the Abstract the authors mentioned In the results section, authors reported many p values! That is far too many for the scope of this study and the authors almost certainly committed type 1 error, as the number of p values is as big as the sample.

What grip strength was measured? Is it relevant to the aim of the study?

What scale did select to measure functional tests in this study? Does “performed or not performed” is a suitable scale to measure this variable?

Is the level and history of athletes considered in sample selection? How authors control this variable?

Althoutgh, this investigation is a poilot, but the sample size is needed to to be sufficient for this research based on power, alpha level and ….? Is the sample size sufficient for this research?

In the manuscript, which results support the discussion on “Our study suggested that kinesiophobia is a significant cause of CLBP among…”

Was this study a cause-effect study or not?

With regards to results, kinesiophobia is the most important factor in CLBP? What about other measured variables?

I also recommend a language check by a native speaker since there are errors all over the text

6. PLOS authors have the option to publish the peer review history of their article (what does this mean?). If published, this will include your full peer review and any attached files.

Reviewer #1: **Yes: **Fatemeh Alirezaei Noghondar

---

## [Author Response · Author response to Decision Letter 0]

28 Jun 2022

June 14 2022

Professor Dr Matias Noll

Academic Editor

PLOS ONE

Dear Professor:

Revised manuscript ID PONE-D-21-25318: Factors affecting chronic low back pain among high school baseball players in Japan: a pilot study

We are grateful for the opportunity to revise our manuscript and for the helpful comments from the reviewers. The comments were highly insightful and enabled us to improve the quality of our paper. Please find attached the point-by-point responses to the reviewers’ comments.

We hope that our revisions have made the manuscript suitable for publication in PLOS ONE

Sincerely,

Hidetoshi Nakao

Faculty of Social Work Studies

Department of Physical Therapy

Josai International University

1 Gumyo, Togane City 

Chiba 283-8555, Japan

Tel.: +81-475-55-8800

Fax: +81-475-55-8811

Email: h_nakao@jiu.ac.jp

 

Authors did a study on Factors Affecting Chronic Low Back Pain Among High School Baseball Players in Japan: A Pilot Study.

Comments from Reviewer 1:

However, I think the study contains many factual errors and needs to be substantially revised to be considered for publication. 

The first concern I have, regards the language use. I think the study needs substantial English language editing to make it more clear for the readers. Secondly, the authors need to tell us about the importance of their study and findings they reached. 

Introduction 

Response:

Thank you for the insightful comments. The revised manuscript has been edited for English language by Editage, a professional English editing service. I have attached the proofreading certificate at the end of this document. Furthermore, significant changes have been made in the revised manuscript (revised text is represented in red) to highlight the importance of this study and conclusions.

The authors start introduction with the sentence IN LINE 45 : „ whereas chronic LBP (CLBP), defined as LBP lasting at least three months, is observed in 1–40% of all baseball players.“ it is needed to be rechecked.

Response:

Thank you for your helpful comment. The age range of baseball players with chronic LBP is wide. In the revised manuscript, we have segregated the LBP prevalence according to age groups.

The revised text is as follows (Line number 47–50 and page number 4):

LBP occurs not only in adulthood but also among young athletes [1]. The LBP prevalence among young baseball players of 12�15.5 years of age ranges from 8.3�15% [2]. Furthermore, LBP affects up to 48% of Japanese college baseball players [3], and the prevalence is also high in older baseball players.

Some sentences need to be more clear for the reader such as in line 51: In addition, reports have identified psychosocial factors and central sensitization as risk factors for CLBP [3, 4]; what does mean…

Sometimes authors present us with statements that need to be followed with references, but the references are omitted. For example, on page …: 

The mental aspects of chronic pain

50 include depression, anxiety, catastrophizing, kinesiophobia, and a decline in self efficacy. What literature are the authors referring to? 

Response:

Thank you for the comment. The text in the revised manuscript was edited to improve clarity. Furthermore, we have included citations in the text to improve clarity on the references. 

The revised text is as follows (Line number 64–68 and page number 5):

Chronic pain constitutes both the physical and mental aspects; the mental aspects include depression, anxiety, catastrophizing, kinesiophobia, and a decline in self-efficacy [10-14]. In addition, reports have identified psychosocial factors and central sensitization as risk factors of CLBP [15, 16]. Psychosocial factors may hinder the athlete from resuming the sport and affect their performance.

The introduction section more or less provides hints to the relevant background but I cannot see how the experimental questions can be drawn from the given information. For instance, in line 59 the statement of the problem has finished with this sentence, Preventing or rehabilitating LBP in high school players may reduce LBP prevalence among college and professional athletes”

The hypotheses cannot be drawn from the information provided? I think it is needed to be revised in a standard manner, because it is not the aim of the study.

Response:

Thank you for the comment. We agree with you. Hence, in the revised manuscript, we have included a paragraph that illustrates the hypothesis and motivation behind this study. Although we have revised the entire Introduction section, the reason for conducting this study is mentioned in the last paragraph of this section. 

The revised text is as follows (Line number 69–73 and page number 5):

To the best of our knowledge, a study examining the association of body compositions and CLBP intensity in high school baseball players is lacking. Hence, this study aimed to examine the factors, such as physical composition, pain evaluation, and load related to LBP, that contribute to CLBP among high school baseball players. We also aimed to identify factors that promote and alleviate CLBP.

METHODS 

We see very important inconsistency: in the Abstract the authors mentioned In the results section, authors reported many p values! That is far too many for the scope of this study and the authors almost certainly committed type 1 error, as the number of p values is as big as the sample. 

Response:

Thank you for the comment. We agree with you. We have reduced the number of p values reported as this study aimed on assessing the physical composition and pain intensity associated with chronic LBP, and identifying the baseball-related loads associated with chronic LBP. The main objective of this study is listed in the last paragraph of the Introduction section of the revised manuscript.

The revised text is as follows (Line number 69–73 and page number 5):

To the best of our knowledge, a study examining the association of body compositions and CLBP intensity in high school baseball players is lacking. Hence, this study aimed to examine the factors, such as physical composition, pain evaluation, and load related to LBP, that contribute to CLBP among high school baseball players. We also aimed to identify factors that promote and alleviate CLBP.

What grip strength was measured? Is it relevant to the aim of the study?

Response:

Thank you for the comment. We agree that there is no relationship between the grip strength and LBP. Therefore, we have removed all information pertaining to grip strength in this study in the revised manuscript. 

What scale did select to measure functional tests in this study? Does “performed or not performed” is a suitable scale to measure this variable?

Response:

Thank you for the comment. In the revised manuscript, we have removed the functional test because it was less important than the other variables.

Is the level and history of athletes considered in sample selection? How authors control this variable?

Response:

Thank you for the comment. Table 1 displays the number of years of experience and position of the athletes at baseline at the onset of LBP. The number of years of experience of players in each group was similar. We recognize that there were many pitchers in the CLBP group; however, since the players’ positions were diverse, we opine that it will be the participants of analysis as the number of data increases in the future. The players studied were all baseball players from the same high school; therefore, we added the amount of practice and frequency of participants as follows. 

The revised text is as follows (Line number 85–89 and page number 6):

The practice frequency of players was six days a week with an average of 4–5 hours per day. Japanese high school baseball games are conducted during the off-season period between December and March, and there are no external games. The practice activities during this period are running, defensive and batting practices, and strength training, and the number of pitching practice decreases.

Althoutgh, this investigation is a poilot, but the sample size is needed to to be sufficient for this research based on power, alpha level and ….? Is the sample size sufficient for this research?

Response:

Thank you for the comment. This study increased the number of participants during the recruitment period. Although the number of participants with chronic LBP did not change, we calculated the effective sample size using the TSK value as an example using the SPSS software. We also provided references on the number of valid pilot studies (line number 141–149 and page number 9–10).

In the manuscript, which results support the discussion on “Our study suggested that kinesiophobia is a significant cause of CLBP among…”

Was this study a cause-effect study or not? 

With regards to results, kinesiophobia is the most important factor in CLBP? What about other measured variables?

Response:

Thank you for the comment. We observed that TSK showed an increase in the longitudinal comparison of athletes with chronic LBP. Other variables either remained the same or reduced, making TSK a CLBP-related factor. Therefore, we examined related factor longitudinal rather than cause-effect.

I also recommend a language check by a native speaker since there are errors all over the text

Response:

As per your suggestion, we have submitted our manuscript to Editage for professional English language editing.

---

## [Decision Letter · Decision Letter 1]

14 Nov 2022

PONE-D-21-25318R1Factors Affecting Chronic Low Back Pain Among High School Baseball Players in Japan: A Pilot StudyPLOS ONE

Dear Dr. Nakao,

Thank you for submitting your manuscript to PLOS ONE. After careful consideration, we feel that it has merit but does not fully meet PLOS ONE’s publication criteria as it currently stands. Therefore, we invite you to submit a revised version of the manuscript that addresses the points raised during the review process.

We look forward to receiving your revised manuscript.

Kind regards,

Yaodong Gu

Academic Editor

PLOS ONE

Additional Editor Comments:

The main purpose of this study shall be clearly described.

Reviewers' comments:

Reviewer's Responses to Questions

**Comments to the Author**

1. If the authors have adequately addressed your comments raised in a previous round of review and you feel that this manuscript is now acceptable for publication, you may indicate that here to bypass the “Comments to the Author” section, enter your conflict of interest statement in the “Confidential to Editor” section, and submit your "Accept" recommendation.

Reviewer #2: (No Response)

Reviewer #3: All comments have been addressed

2. Is the manuscript technically sound, and do the data support the conclusions?

Reviewer #2: No

Reviewer #3: Yes

3. Has the statistical analysis been performed appropriately and rigorously? 

Reviewer #2: I Don't Know

Reviewer #3: Yes

4. Have the authors made all data underlying the findings in their manuscript fully available?

Reviewer #2: Yes

Reviewer #3: Yes

5. Is the manuscript presented in an intelligible fashion and written in standard English?

Reviewer #2: No

Reviewer #3: Yes

6. Review Comments to the Author

Reviewer #2: This article is a polit study, but only briefly describes the purpose of this study can help implement corrective measures for improving LBP. The significance of this study is not specified in detail

In the article, which results support the discussion about kinesiophobia is an important cause of CLBP.

The conclusion is too simple and needs to be further summarized

Reviewer #3: Review comment

This manuscript entitled “Factors Affecting Chronic Low Back Pain Among High School Baseball Players in Japan: A Pilot Study” primarily propose to examined the factors associated with CLBP among high school baseball players. But there are only few questions should be addressed before this manuscript can be accepted for publication. You can revise this paper more properly. I suggest that you improve the description below.

Specific comments

1.At the end of the abstract section, the hypothesis for this study seems to be missing. Please add this section to the manuscript.

2.A description of the application of the results in this study seems to be missing in the discussion and conclusion section. In fact this is necessary to enhance the quality and clarify the significance of this study.

3.In fact LBP is not only present in the baseball player community, but also in the table tennis and tennis player communities. Therefore, add description content into discussion section on LBP of racket sports is necessary. This would enhance the perspective of this study.

7. PLOS authors have the option to publish the peer review history of their article (what does this mean?). If published, this will include your full peer review and any attached files.

Reviewer #2: No

Reviewer #3: **Yes: **Yuqi He

---

## [Author Response · Author response to Decision Letter 1]

20 Nov 2022

Authors did a study on Factors Affecting Chronic Low Back Pain Among High School Baseball Players in Japan: A Pilot Study.

Comments from Reviewer 2: 

This article is a polit study, but only briefly describes the purpose of this study can help implement corrective measures for improving LBP. The significance of this study is not specified in detail

In the article, which results support the discussion about kinesiophobia is an important cause of CLBP.

The conclusion is too simple and needs to be further summarized

Response:

Thank you very much for reviewing our manuscript and offering valuable advice. We have addressed your comments with point by point response, and revised the manuscript accordingly. 

We agree with your observation. I have added extra information to the discussion section, specifically focusing on the association of kinesiophobia with CLBP and the importance of managing TSK score during exersice for CLBP.

The revised text is as follows (Lines 241–249):

Thus, kinesiophobia affects joint movement ability; therefore, continuing training with running and squats poses a risk. However, aerobic exercises auch as walking and strength execises specific to the thoracic, lumbar, and posterior hip regions have been reported to be effective in imporoving CLBP. Therefore, an exercise program with a controlled TSK score is recommended for atheletes with CLBP.

Our present findings suggest that higher TSK in CLBP is seen among high school baseball players with lumbar spondylolysis. Future studies should investigate the association between kinesiophobia and spondylolysis in other sports, such as racquet and contact sports, as they may also have high TSK scores and lumbar spondylolysis rates in CLBP.

We agree with you regarding your suggestion that the conclusion is simple, and we have made significant revisions. The revised text is as follows. (Lines 256–262)

Our study aimed to examine factors, such as physical composition, pain evaluation, and load related to LBP, that contribute to CLBP among high school baseball players. We identified kinesiophobia and lumbar spondylolysis as factors related to CLBP. Therefore, particular attention should be given to kinesiophobia during running and barbell squat training, especially in young athletes. Furthermore, the TSK score and presence of lumbar spondylolysis should be considerd when creating an exercise program for high school baseball players with CLBP. 

Comments from Reviewer 3:

This manuscript entitled “Factors Affecting Chronic Low Back Pain Among High School Baseball Players in Japan: A Pilot Study” primarily propose to examined the factors associated with CLBP among high school baseball players. But there are only few questions should be addressed before this manuscript can be accepted for publication. You can revise this paper more properly. I suggest that you improve the description below.

Dear Reviewer 

　

Thank you very much for reviewing our manuscript and offering valuable advice. We have addressed your comments with point by point response, and revised the manuscript accordingly. 

Specific comments

1.At the end of the abstract section, the hypothesis for this study seems to be missing. Please add this section to the manuscript.

Response:

Thank you for your helpful comment. Accordingly, the following sentence was inserted at the end of the abstract section. 

The revised text is as follows (Lines 42–44):

 Kinesiophobia and the presence of lumbar spondylolysis should be considerd when creating an exercise program for high school baseball players with CLBP.

2.A description of the application of the results in this study seems to be missing in the discussion and conclusion section. In fact this is necessary to enhance the quality and clarify the significance of this study.

Response:

Thank you for the insightful comment. I have added the discussion regarding the necessity of TSK scores assessment for athletes with CLBP for continuation of exercise.

The revised text is as follows (Lines 241–245):

Thus, kinesiophobia affects joint movement ability; therefore, continuing training with running and squats poses a risk. However, aerobic exercises such as walking [35] and strength training specific to the thoracic, lumbars, and posterior hip regions have been reported to be effective in imporoving CLBP [36]. Therefore, an exercise program with a controlled TSK score is likely recommended for atheletes with CLBP.

3.In fact LBP is not only present in the baseball player community, but also in the table tennis and tennis player communities. Therefore, add description content into discussion section on LBP of racket sports is necessary. This would enhance the perspective of this study.

Response:

Thank you for your helpful comment. We agree with your observation. We have added relevant information to the discussion section regarding the need to pursue the assessment of association of kinesiophobia and lumbar spondylolysis with CLBP in other sports as a topic for future study.

The revised text is as follows (Lines 246–249):

Our present findings suggest that higher TSK in CLBP is seen among high school baseball players with lumbar spondylolysis. Future studies should investigate the association between kinesiophobia and spondylolysis in other sports, such as racquet and contact sports, as they may also have higher TSK scores and lubar spondylolysis rates in CLBP.

---

## [Decision Letter · Decision Letter 2]

2 Jan 2023

Factors Affecting Chronic Low Back Pain Among High School Baseball Players in Japan: A Pilot Study

PONE-D-21-25318R2

Dear Dr. Nakao,

We’re pleased to inform you that your manuscript has been judged scientifically suitable for publication and will be formally accepted for publication once it meets all outstanding technical requirements.

Kind regards,

Yaodong Gu

Academic Editor

PLOS ONE

Additional Editor Comments (optional):

N/A

Reviewers' comments:

Reviewer's Responses to Questions

**Comments to the Author**

1. If the authors have adequately addressed your comments raised in a previous round of review and you feel that this manuscript is now acceptable for publication, you may indicate that here to bypass the “Comments to the Author” section, enter your conflict of interest statement in the “Confidential to Editor” section, and submit your "Accept" recommendation.

Reviewer #2: All comments have been addressed

Reviewer #3: All comments have been addressed

2. Is the manuscript technically sound, and do the data support the conclusions?

Reviewer #2: Yes

Reviewer #3: Yes

3. Has the statistical analysis been performed appropriately and rigorously? 

Reviewer #2: Yes

Reviewer #3: Yes

4. Have the authors made all data underlying the findings in their manuscript fully available?

Reviewer #2: Yes

Reviewer #3: Yes

5. Is the manuscript presented in an intelligible fashion and written in standard English?

Reviewer #2: Yes

Reviewer #3: Yes

6. Review Comments to the Author

Reviewer #2: (No Response)

Reviewer #3: Thank you for author answerd all the comments in an appropriately way. In my point of view, this article could be accept to publish.

7. PLOS authors have the option to publish the peer review history of their article (what does this mean?). If published, this will include your full peer review and any attached files.

Reviewer #2: No

Reviewer #3: **Yes: **Yuqi He

---

## [Editor Report · Acceptance letter]

17 Jan 2023

PONE-D-21-25318R2 

Factors affecting chronic low back pain among high school baseball players in Japan: a pilot study 

Dear Dr. Nakao:

I'm pleased to inform you that your manuscript has been deemed suitable for publication in PLOS ONE. Congratulations! Your manuscript is now with our production department. 

Kind regards, 

on behalf of

Professor Yaodong Gu 

Academic Editor

PLOS ONE